# A Multi-Center Study of *BRCA1* and *BRCA2* Germline Mutations in Mexican-Mestizo Breast Cancer Families Reveals Mutations Unreported in Latin American Population

**DOI:** 10.3390/cancers11091246

**Published:** 2019-08-26

**Authors:** Oliver Millan Catalan, Alma D. Campos-Parra, Rafael Vázquez-Romo, David Cantú de León, Nadia Jacobo-Herrera, Fermín Morales-González, César López-Camarillo, Mauricio Rodríguez-Dorantes, Eduardo López-Urrutia, Carlos Pérez-Plasencia

**Affiliations:** 1Laboratorio de Genómica, Instituto Nacional de Cancerología (INCan). Av. San Fernando 22, Col. Sección XVI, C.P. Tlalpan, Ciudad de México 14080, Mexico; 2Departamento de Cirugía de Tumores Mamarios, Instituto Nacional de Cancerología (INCan), Av. San Fernando 22, Col. Sección XVI, C.P. Tlalpan, Ciudad de México 14080, Mexico; 3Dirección de Investigación, Instituto Nacional de Cancerología (INCan), Av. San Fernando 22, Col. Sección XVI, C.P. Tlalpan, Ciudad de México14080, Mexico; 4Unidad de Bioquímica, Instituto Nacional de Ciencias Médicas y Nutrición, Salvador Zubirán (INCMNSZ), Av. Vasco de Quiroga 15, Col Belisario Dominguez. C.P. Tlalpan, Ciudad de México 14080, Mexico; 5Instituto Jalisciense de Cancerología. Coronel Calderón 715, Guadalajara 44280, Jalisco, Mexico; 6Posgrado en Ciencias Genómicas, Universidad Autónoma de la Ciudad de México, San Lorenzo 290, Del Valle Sur, Benito Juarez, Ciudad de México 03100, Mexico; 7Instituto Nacional de Medicina Genómica, Ciudad de México 14610, Mexico; 8Laboratorio de Genómica, Unidad de Biomedicina, FES-IZTACALA, UNAM, Tlalnepantla 54090, Mexico

**Keywords:** breast cancer, *BRCA1*, *BRCA2*, NGS, Latin American population, Mexican-mestizo population

## Abstract

The presence of germline and somatic deleterious mutations in the *BRCA1* and *BRCA2* genes has important clinical consequences for breast cancer (BC) patients. Analysis of the mutational status in *BRCA* genes is not yet common in public Latin American institutions; thus, our objective was to implement high-performance technology with highly reliable results with the possibility of analyzing several patients simultaneously, therefore reducing cost and work time. A prospective cohort of 252 unrelated sporadic breast cancer patients from the Mexican-mestizo population were analyzed using next generation sequencing (NGS) based on ion semiconductor sequencing. We found 28 pathogenic mutations (25 in *BRCA1* and 13 in *BRCA2*), 11 of which had not been reported previously in Hispanic or Latin American populations. A total of 38 patients were positive for a pathogenic mutation representing 15% of our Mexican women cohort with breast cancer; 25 for *BRCA1*; and 13 for *BRCA2*. Our results revealed that there are mutations not analyzed by mutations panels, and our findings support the suitability of massive sequencing approaches in the public institutions of developing countries. Hence, *BRCA* screening should be offered to patients with breast cancer regardless of their family history of cancer in order to identify unaffected family carriers.

## 1. Introduction

In spite of the time and resources devoted to its study, breast cancer (BC) remains the most diagnosed neoplasm and has the third highest number of reported deaths worldwide [1]. Most breast cancer cases are sporadic; however, approximately 5% of breast cancers are attributable to germline *BRCA1* or *BRCA2* mutations. Although different studies have shown that the penetrance of deleterious *BRCA* is variable, for *BRCA1* mutation carriers, the average cumulative risk of developing BC by age 70 has been estimated at 65%, and for *BRCA2* mutation carriers, at 45% [2]. Currently, there are more than 3200 described mutations that confer breast or ovarian cancer susceptibility [3]. Rather than being evenly distributed among the population, recent research has shown that *BRCA* mutations are distributed geographically and that these mutation subsets contain important numbers of exclusive mutations [4].

Therefore, it is important to search for *BRCA* mutations as deeply as possible in populations as diverse as possible. Achieving this is relatively easier in developed countries with efficient screening policies [5]; yet, in countries where widespread public *BRCA* mutation screening programs for high-risk patients are yet to be fully implemented, more efficient methods need to be pursued. Recently, parallel next-generation sequencing has proven to be a feasible option for *BRCA* mutation detection in settings such as Tunisia [6], Brazil [7], and China [8], among others.

In Mexico, available evidence strongly suggests that deleterious *BRCA* mutations are directly related to a higher proportion of breast cancers in the Mexican-mestizo population, such as an earlier age at onset of breast cancer in Mexican women (<50 years) [9,10], a high prevalence (23%) of triple negative breast cancer (TNBC) [11], and a high proportion (20%) of TNBC patients that carry *BRCA* mutations [12]. Hence, there is a need to continue the previous efforts [11,13] toward a comprehensive *BRCA* mutation catalog of the Mexican-mestizo population that can bolster current BC diagnosis and treatment resources.

The purpose of this study was to analyze *BRCA* mutation frequency in a Mexican, multi-center cohort of breast and ovarian cancer patients through ion semiconductor sequencing.

## 2. Results

The screened patients ranged from ages 28 to 63, with a mean age of 40 (*n* = 252); most samples belonged to stage III (38%, *n* = 95), and stages I–IV were all represented (Table 1). From a histological perspective, ductal carcinomas were far more frequent than lobular or mixed tumors (Table 2). 

The raw sequencing files obtained from this study can be consulted at http://www.ncbi.nlm.nih.gov/bioproject/559341. We found a total of 95 variants in the exons and adjacent intronic regions of the *BRCA1* and *BRCA2* genes; 40 of these were located in *BRCA1* and 55 in *BRCA2* (data not shown). From these variants, 28 were reported as pathogenic in the ClinVar database or as Class 5 in the BIC database, 16 in *BRCA1*, and 12 in *BRCA2*. From these, 17 had been previously reported in Latin American in Hispanic populations and 11 had not. The most common *BRCA1* mutation was c.66_7delAG, present in four samples, while there were 11 mutations with only one report. As for *BRCA2*, the most common mutation was c.1813delA, which was present in two samples (Table 3). A total of 38 patients were positive for a pathogenic mutation, representing 15% of our Mexican women cohort with breast cancer: 25 for *BRCA1* and 13 for *BRCA2*. Mutation distribution was uneven among molecular subtypes: 20% of the triple negative (TNBC) samples harbored pathogenic mutations in *BRCA1* compared to 6.6% and 8% of the HR+ and Her2−, and Her2+ samples, respectively. Pathogenic mutations in *BRCA2* were only present in HR+ and Her2−, and Her2+ tumors (Table 4).

The 28 pathogenic variants comprised mostly frameshift mutations, with a smaller number of amino acid substitutions and only two splicing mutations present (Table 5). Both frameshift and amino acid substitutions were plotted along the BRCA 1 and 2 domain structure to find out whether mutations were more frequent in a particular region of either protein. Figure 1 shows that the distribution is somewhat uniform along the length of both proteins, although there is a higher number of mutations in the coiled coil domain of BRCA 1 and the BRC repeats of BRCA 2.

## 3. Discussion

At present, there are few studies characterizing germline mutations in Mexican-mestizo breast cancer patients. The main disadvantage of these studies is that the methods employed do not analyze the complete sequence of *BRCA* genes; rather, they search for a pre-defined set of mutations [13,34,35,36]. An example is the panel known as HISPANEL, which analyzes 114 frequent mutations in data obtained in the US with Latin or Hispanic patients. HISPANEL has been used to characterize mutations in Mexican BC patients [32,36]; however, by analyzing only 114 frequent mutations, it is not possible to identify pathogenic mutations that are under-represented in the population. In this study, we reported 11 mutations that had not been analyzed in Latino or Hispanic populations before, which further evinces the advantages of analyzing genes associated with cancer development through next generation sequencing (NGS) technologies.

In Mexico, access to genetic tests to identify mutations in genes involved in cancer development is still limited. Scientific research projects are providing important data to awaken the interest of policy makers in government to mitigate this problem. In the present work, we screened for germline mutations from a wide area of Mexican territories by recruiting 252 patients from two centers that concentrate patients from most of the Mexican states. The design of the present study allowed us to obtain a cohort of BC patients that reflected the genetic diversity of the Mexican population. Thus, we were able to find 95 variants: 28 of them were classified as pathogenic and 11 had not been reported before in Latin American or Hispanic populations. A recent study by Fernández-López shows that the distribution of *BRCA* variants in the Mexican population may not differ significantly in different regions of the country. Although the *BRCA* variants obtained in this work came from a large number of healthy subjects [17], we still found pathogenic variants previously unidentified in the Mexican population.

Despite the fact that our patient cohort was not selected by familial history, the proportion of BC samples where we found *BRCA* mutations was close to the recently reported values in hereditary breast cancer patients from Colombia [33], Spain [15], Saudi Arabia [37], China [38], and Israel [39], among others. However, this was different from Brazil—a Latin American country often compared to Mexico in several aspects—where the *BRCA* mutation frequency among BC cancer patients is under 4% [40], and Iran, where studies point at a *BRCA* mutation frequency of 31% among BC patients [41].

The high proportion of TNBC samples matches the result from a previous single-center study that included samples largely from the central region of Mexico, where a 23% of TNBC was found as well [11]. This proportion is similar to global estimates [42], except for some Asian countries where the proportion is around 15% [43]. Likewise, *BRCA* mutations in TNBC samples were comparable to the previously observed 23% in patients from Mexico City [36]. This proportion is higher than that found in unselected TNBC, which is around 19% [12], but can be lower in regions such as Australia, where a 9.3% is reported [44]. Interestingly, we found only *BRCA1* mutations in these samples, so we support the idea laid out by Baretta and colleagues [45] that the prognostic value of *BRCA1* and *BRCA2* mutations should be accounted for separately, at least regarding TNBC.

## 4. Materials and Methods

### 4.1. Patient Cohort

A total of 252 unrelated patients were recruited for screening between January 2015 and November 2017: 84 BC patients from the Oncologic Hospital of Guadalajara (Guadalajara City, Mexico) and 168 BC patients from the Instituto Nacional de Cancerología (Mexico City, Mexico). Both centers concentrate patients from the neighboring states, i.e., the north and northwestern (Oncologic Hospital of Guadalajara) and center, south, and southeastern (Instituto Nacional de Cancerología) regions of Mexico, widening the coverage of our study. All patients included in the present study signed an informed consent form.

### 4.2. DNA Isolation

DNA was extracted from peripheral blood with the QIAamp DNA Blood Mini kit following the manufacturer’s instructions (Qiagen, cat. no. 51106, Hilden, Germany). DNA integrity was verified by agarose electrophoresis and the concentration was determined using RNase P Detection Reagent (FAM) (Applied Biosystems, cat. no. 4316831).

### 4.3. Ion Semiconductor Sequencing

*BRCA1* and *BRCA2* were amplified using the Ion Ampliseq *BRCA 1* and *2* panel (Thermo Fisher Scientific). This panel includes 167 primers pairs in three pools. For library preparation, we used 25 ng of DNA, and amplifications of each patient were marked with a unique Ion Xpress Barcode Adapter (Thermo Fisher Scientific cat. no. 4471250), purified with AMPure XP reagent (Beckman Coulter cat. no. A63881), and quantified with an Ion Library Taqman Quantitation Kit (Thermo Fisher Scientific cat. no. 4468802). Libraries were mixed in equimolar concentrations prior to emulsion PCR and Ion Sphere particle enrichment with a Hi-Q OT2 Reagent Kit (Thermo Fisher Scientific cat. no. A27743) in the Ion OneTouch 2 System (Thermo Fisher Scientific). For sequencing, we used the Ion PGM Hi-Q Sequencing kit (REFA25589) with chips 314, 316, or 318 in the Ion torrent PGM (Personal Genome Machine) instrument (Thermo Fisher Scientific).

### 4.4. Data Analysis

The sequences data were aligned to the hg19 human reference genome (GRCh37). The .bam files were exported to the Ion Reporter mutation analysis.

## 5. Conclusions

Our study allowed us to analyze mutations not represented in previous screenings of Mexican BC patients, identifying pathogenic mutations in a wide sample of patients from different states of the Mexican territory. Finding unrepresented mutations in Hispanic or Latino populations demonstrates that sequencing, rather than screening for a pre-defined set of mutations, is a more robust approach to the analysis and characterization of pathogenic variants in genes associated with cancer development.

## Figures and Tables

**Figure 1 cancers-11-01246-f001:**
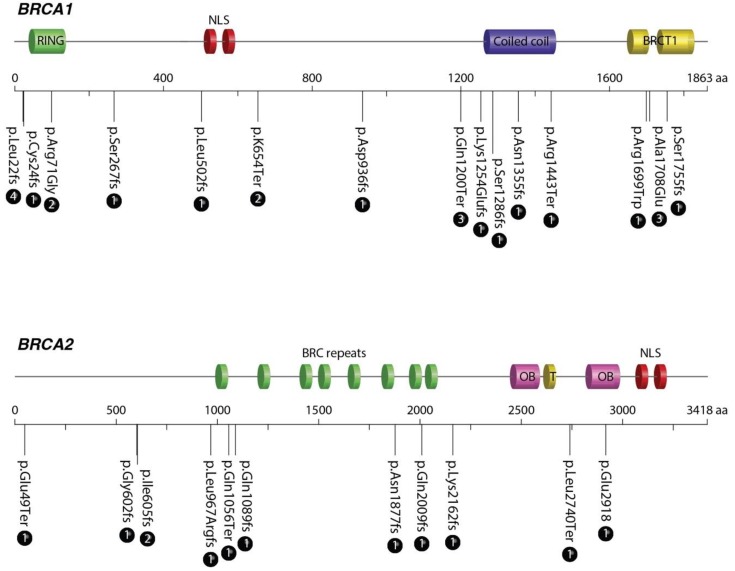
Schematic representation of the *BRCA* proteins showing the locations of the pathogenic mutations found. The mutations are represented by the amino acid substitutions or the presence of a frameshift (fs). Numbers in circles represent the frequency of each mutation.

**Table 1 cancers-11-01246-t001:** *BRCA* mutations found in 252 unrelated breast cancer (BC) patients, divided by stage.

Stage	Percentage	Frequency
I	13.5%	34
II	29.7%	75
III	38%	95
IV	18.8%	48

**Table 2 cancers-11-01246-t002:** *BRCA* mutations found in 252 unrelated BC patients, divided by histology.

Histology	Percentage	Frequency
Ductal	83.7%	211
Lobular	11.1%	28
Mixed	5.2%	13

**Table 3 cancers-11-01246-t003:** *BRCA* pathogenic mutations found in 252 unrelated BC patients.

Gene	Coding	Amino Acid Change	dbSNP	ClinVar/BIC Category	Frequency (Age of Diagnosis)	Ref. in LATAM Hispanic Population
*BRCA1*	c.66_67delAG	p.Leu22fs	rs80357783	Pathogenic/Class 5	4 (35, 36, 40, 44)	[14] [15] [16]
	c.69_70insAG	p.Cys24fs	rs80357914	Pathogenic/Class 5	1 (42)	[17]
	c.211A>G	p.Arg71Gly	rs80357382	Pathogenic/Pending	2 (49, 49)	[15] [17] [18]
	c.798_799delTT	p.Ser267fs	rs80357724	Pathogenic/Class 5	1 (39)	Not reported
	c.1504_1508delTTAAA	p.Leu502fs	rs876659139	Pathogenic/Class 5	1 (36)	Not reported
	c.1960A>T	p.K654Ter	rs80357355	Pathogenic/Class 5	2 (32, 50)	[16] [19]
	c.2806_2809delGATA	p.Asp936fs	rs80357832	Pathogenic/Class 5	1 (45)	Not reported
	c.3598C>T	p.Gln1200Ter	rs62625307	Pathogenic/Class 5	3 (44, 45, 48)	[20]
	c.3759_3760delTA	p.Lys1254Glufs	rs80357520	Pathogenic/Class 5	1 (37)	Not reported
	c.3858_3861delTGAG	p.Ser1286fs	rs80357842	Pathogenic/Class 5	1 (34)	[21]
	c.4065_4068delTCAA	p.Asn1355fs	rs80357508	Pathogenic/Class 5	1 (40)	Not reported
	c.4327C>T	p.Arg1443Ter	rs41293455	Pathogenic/Class 5	1 (41)	[17] [19]
	c.5095C>T	p.Arg1699Trp	rs55770810	Pathogenic/Pending	1 (46)	[22]
	c.5123C>A	p.Ala1708Glu	rs28897696	Pathogenic/Pending	3 (40, 41, 44)	[23] [24] [25] [26]
	c.5263_5264insC	p.Ser1755fs	rs80357906	Pathogenic/Class 5	1 (50)	[27]
	IVS5+1G>A c.212+1G>A	Splicing mutation	rs80358042	Pathogenic/Pending	1 (35)	[28]
*BRCA2*	c.145G>T	p.Glu49Ter	rs80358435	Pathogenic/Class 5	1 (31)	[28] [29]
	c.1806insA	p.Gly602fs	rs80359307	Pathogenic/Class 5	1 (37)	Not reported
	c.1813delA	p.Ile605fs	rs80359306	Pathogenic/Class 5	2 (33, 46)	Not reported
	c.2899_2900delCT	p.Leu967Argfs	rs80359361	Pathogenic/Class 5	1 (35)	Not reported
	c.3166C>T	p.Gln1056Ter	rs79728106	Pathogenic/Class 5	1 (37)	[30]
	c.3492insT	p.Gln1089fs	rs80359380	Pathogenic/Class 5	1 (40)	[28] [31]
	c.5631delC	p.Asn1877fs	rs397507357	Pathogenic/Class 5	1 (41)	[20]
	c.6244_6244delG			Pathogenic/Class 5	1 (47)	Not reported
	c.6024_6025insG	p.Gln2009fs	rs80359554	Pathogenic /Class 5	1 (40)	[32] [33]
	c.6486_6489delACAA	p.Lys2162fs	rs80359598	Pathogenic/Class 5	1 (31)	[28]
	c.8219T>G	p.Leu2740Ter	rs80359070	Pathogenic/Class 5	1 (33)	Not reported
	c.8754G>A	Splicing mutation p.Glu2918=	rs80359803	Pathogenic, Likely pathogenic/Pending	1 (38)	Not reported

**Table 4 cancers-11-01246-t004:** *BRCA* mutations found in 252 unrelated BC patients, divided by molecular subtype.

	Molecular Subtype		Mutations					
			*BRCA1*		*BRCA2*		TOTAL	
	Percentage	Frequency	Percentage	Frequency	Percentage	Frequency	Percentage	Frequency
HR+, Her2−	48%	121	6.6%	8	4.1%	5	10.7%	13
Her2+	24.7%	62	8%	5	12.9%	8	20.9%	13
Triple Negative	23.8%	60	20%	12	–	–	20%	12
Unknown	3.5%	9	–	–	–	–	–	–

**Table 5 cancers-11-01246-t005:** *BRCA* mutations found in 252 unrelated BC patients, divided by mutation type.

Mutation Type	*BRCA1*	*BRCA2*
Frameshift	8	8
Amino acid substitution	6	3
Splicing mutation	2	1

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
