# Peer review of "A Multi-Center Study of BRCA1 and BRCA2 Germline Mutations in Mexican-Mestizo Breast Cancer Families Reveals Mutations Unreported in Latin American Population"

_cancers, 2019, doi:10.3390/cancers11091246_

Round 1

Reviewer 1 Report

Major issues:

1.    Line 87-89: how do authors define pathogenic variants? Especially for those variants that have not yet reported before?

2.    It is very confusing about the way how to count mutations in the manuscript. On Line 33 in abstract, Thirty-nine (39) pathogenic mutations, however, on Line 88-89, “From these variants, 38 were identified as pathogenic” , and then in Table 3: only 28 unique mutations in 38 patients. Authors should clearly report how many unique mutations have been identified in BRAC1/2 gene. I guess it should be 28.

3.    Same issue as #2, on Line 104-105, “38 pathogenic variants comprised …. (Table 5)”. On Table 5, total count of the mutations is 28.

4.    Are the pathogenic variants germline mutations? Or somatic mutation? In discussion (Line 134-136), authors claim that this study screens germline mutation. However, the manuscript doesn’t report how somatic and germline mutations are classified. No experimental design has suggested that the study only screens germline mutations

5.    Authors need to deposit the sequence data such as bam files to public databases.

6.    Another confusing number. Line 127-128: “In this study, we reported 14 mutations that had not been analyzed in Latino or Hispanic populations before…”. However, Line 138-139: “Hence, we were able to find 1806 variants; 38 of them were classified as pathogenic and 11 had not been reported before in Latin American or Hispanic populations.”  14 or 11, which one is correct one?

7.    The whole result section should be carefully re-edited in order to provide clear and accurate results to readers.

8.    Line 191-196, the conclusion is confusing and subjective as well. Identifying new mutations that haven’t been reported before doesn’t mean those mutations are not represented in breast cancer patients of the Mexican population. NGS provides some advantage for mutation finding. However, the manuscript doesn’t have enough evidence to prove that NGS is the only way to “analyze and characterize pathogenic variants in genes associated to cancer development.”

 Minor issues:

1.    Line 34: missing word:  eleven of the ??

2.    Line50:“developing OC” What is OC? Is it a typo? BC? 

3.    Ion Torrent PGM system,semiconductor sequencing (Ion semiconductor sequencing?). Please use same nomenclature throughout the article for the next-generation sequencing method used in the study.

4.    Line 129: typo:“further evince”

5.    Line 165:“168 BC and patients” Should the word “and” be deleted? 

Reviewer 2 Report

This is a study examining an under-represented population (Mexican) in the BRCA1/2 literature.  The authors examined 252 patients with breast cancer, and report on 39 pathogenic variants in BRCA1/2.  It is a well written manuscript, with tables summarizing their findings in clinically relevant categories. 

It is unclear what patients undergo genetic testing in Mexico (e.g. genetic testing criteria in Mexico).  How do testing criteria in Mexico relate to other jurisdictions such as UK, National Comprehensive Cancer Network.  A description of the 252 patients should also be tabulate (e.g. demographics table).  The age of different mutation carriers is also a useful table.  The classification of the variants should be described (e.g. ACMG annotation, ENIGMA, BRCA exchange, etc).  One classification system should be used (pathogenic, likely pathogenic as per ACMG is recommended). 

Minor points:

Some of the figures use three letter and single letter amino acid short forms.  BRCA1/2 should be italicized, and consistently one term should be used (e.g. BRCA vs BRCA1/2).
